# Cloning and Characterization of a Novel *N*-Acetyl-D-galactosamine-4-*O*-sulfate Sulfatase, SulA1, from a Marine *Arthrobacter* Strain

**DOI:** 10.3390/md22030104

**Published:** 2024-02-23

**Authors:** Monica Daugbjerg Christensen, Leila Allahgholi, Javier A. Linares-Pastén, Ólafur Friðjónsson, Hörður Guðmundsson, Varsha Kale, Roya R. R. Sardari, Guðmundur Ó. Hreggviðsson, Eva Nordberg Karlsson

**Affiliations:** 1Matís, Vínlandsleið 12, 113 Reykjavik, Iceland; olafur@matis.is (Ó.F.); hordurg@matis.is (H.G.); matis@matis.is (V.K.); gudmundo@matis.is (G.Ó.H.); 2Faculty Food Science and Nutrition, University of Iceland, 101 Reykjavik, Iceland; 3Division of Biotechnology, Department of Chemistry, Lund University, P.O. Box 124, 22100 Lund, Sweden; leila.allahgholi@biotek.lu.se (L.A.); javier.linares-pasten@biotek.lu.se (J.A.L.-P.); roya.sardari@biotek.lu.se (R.R.R.S.); eva.nordberg_karlsson@biotek.lu.se (E.N.K.); 4Faculty of Life and Environmental Sciences, University of Iceland, 101 Reykjavik, Iceland

**Keywords:** marine bacterium, chondroitin sulfate, *N*-acetylgalactosamine-4-*O*-sulfate, sulfatase

## Abstract

Sulfation is gaining increased interest due to the role of sulfate in the bioactivity of many polysaccharides of marine origin. Hence, sulfatases, enzymes that control the degree of sulfation, are being more extensively researched. In this work, a novel sulfatase (SulA1) encoded by the gene *sulA1* was characterized. The *sulA1*-gene is located upstream of a chondroitin lyase encoding gene in the genome of the marine Arthrobacter strain (MAT3885). The sulfatase was produced in *Escherichia coli*. Based on the primary sequence, the enzyme is classified under sulfatase family 1 and the two catalytic residues typical of the sulfatase 1 family—Cys57 (post-translationally modified to formyl glycine for function) and His190—were conserved. The enzyme showed increased activity, but not improved stability, in the presence of Ca^2+^, and conserved residues for Ca^2+^ binding were identified (Asp17, Asp18, Asp277, and Asn278) in a structural model of the enzyme. The temperature and pH activity profiles (screened using *p*-nitrocatechol sulfate) were narrow, with an activity optimum at 40–50 °C and a pH optimum at pH 5.5. The *T_m_* was significantly higher (67 °C) than the activity optimum. Desulfation activity was not detected on polymeric substrates, but was found on GalNAc4S, which is a sulfated monomer in the repeated disaccharide unit (GlcA–GalNAc4S) of, e.g., chondroitin sulfate A. The position of the sulA1 gene upstream of a chondroitin lyase gene and combined with the activity on GalNAc4S suggests that there is an involvement of the enzyme in the chondroitin-degrading cascade reaction, which specifically removes sulfate from monomeric GalNAc4S from chondroitin sulfate degradation products.

## 1. Introduction

Chondroitin sulfate/dermatan sulfate sulfatases (CS/DS-sulfatases) are specialized enzymes. They hydrolyze and remove sulfate ester groups from mono-, di-, oligo-, and polysaccharides of chondroitin sulfate (CS) or dermatan sulfate (DS). These enzymes can exhibit either endo- or exo-acting activities. This enables them to either remove sulfate groups located along the poly-/oligosaccharide chain or cleave sulfate groups from the reducing and/or the non-reducing ones [1,2].

The majority of sulfatases, including CS/DS-sulfatases, belong to Type 1 sulfatases, also known as arylsulfatases, or more accurately, formylglycine-dependent sulfatases (FGly-SULFs [3,4]; http://abims.sb-roscoff.fr/sulfatlas/ (accessed on 18 February 2024)). They are given this name due to the post-translational formation of the active site residue Cα-formylglycine (FGly), derived from a conserved cysteine (or serine) residue found in the signature motif C/S-X-P-X-R [5]. CS/DS-sulfatases have been further categorized based on their substrate specificity, namely Δ^4,5^hexuronate-2-*O*-sulfatase, *N*-acetylgalactosamine-4-*O*-sulfatase (GalNAc-4-*O*-sulfatase), and *N*-acetylgalactosamine-6-*O*-sulfatase (GalNAc-6-*O*-sulfatase). These sulfatases target specific sulfate groups within CS/DS molecules, enabling the modification and degradation of these biomolecules [6,7]. However, despite the discovery of new enzymes and a large amount of genomic data, sulfatases remain, to a large extent, underexplored concerning substrate specificity and kinetic properties [8,9]. So far, only three Δ^4,5^hexuronate-2-*O*-sulfatase [10,11,12], three endo-type and two exo-type GalNAc-4-*O*-sulfatases [2,7,12,13,14], and three exo-type GalNAc-6-*O*-sulfatases [7,12,14], have been identified and characterized.

CS and DS are types of glycosaminoglycans (GAG) present in the cartilage tissue of both land and marine animals [15]. Their overall structures are characterized by repeated disaccharide units consisting of the glucuronic acid (GlcA) and *N*-acetyl-galactosamine (GalNAc) or L-iduronic acid (IdoUA) and GalNAc, respectively, linked by β-1,4-glycosidic bonds. These polysaccharides can be classified into different classes based on their heterogeneous length and degree of sulfation, and are influenced by the species of origin of the CS [16]. The two major classes of CS are CS-A and CS-C, which are distinguished by the position of the sulfate group on the GalNAc unit. In CS-A, the sulfate group is located at C4, while, in CS-C, it is positioned at C6 on the GalNAc molecule. However, more distinct and complex CS structures exist, like CS-AC, characterized by alternating C4 and C6 sulfation on the GalNAc unit and fucosylated CS from sea cucumber [17]. CS is widely recognized for its functional properties and diverse bioactivities, including anti-inflammatory properties [18,19], anti-cancer properties [20], anti-coagulating properties [21], and it is used as osteoarthritis treatment [22]. These characteristics make CS molecules intriguing for pharmaceutical and nutraceutical applications. The bioactivities of CS are primarily attributed to its sulfate content, the positions of the sulfate groups, and the size of the polymer.

In the recent years, researchers at Matís successfully isolated and sequenced the marine chondroitin-degrading *Arthrobacter* strain MAT3885. This discovery was made using microbial traps coated with CS [23]. The gene encoding the sulfatase described here, designated SulA1, originates from this strain and is classified under the Type 1 sulfatase family based on sequence characteristics. The substrate specificity was investigated and specific activity on a natural substrate was only found on monosulfated GalNAc4S, which is a building block in both CS and DS. Modelling of the structure and substrate docking revealed that binding of GalNAc4S was energetically favored, while the position of the substrate molecule was somewhat unexpected in relation to the calcium ligand bound in the active site. This study contributes insights into the substrate specificity of one of the relatively underexplored sulfatases from marine bacteria. Furthermore, these findings have the potential to serve as a valuable resource for further investigations into the structural features and practical applications of CS sulfatases.

## 2. Results

### 2.1. Phylogenetic Identification, Identification of SulA1, and Modelling of the Enzyme

A BlastN analysis of the MAT3885 16S rDNA sequence confirmed its affiliation with *Arthrobacter*. The closest relative was found to be *Arthrobacter cryoconiti* strain GS9 (99.79% id). Potential sulfatase genes were observed in the vicinity of the chondroitin lyase gene, ChoA1. ChoA1 belongs to the polysaccharide lyase family 8, as described by Kale et al. [23]. This family of lyases, known to act on CS with sulfation on both C4 and C6, is referred to as CS-AC (www.cazy.org (accessed on 18 February 2024)). A small chondroitin lyase gene cluster including two potential sulfatase-encoding genes (*sulA1* and *sulA2*) was identified (Figure 1). The *sulA1* gene was located immediately upstream of the chondroitin lyase gene, *choA1*, making it an interesting target for further investigation. The SulA1 sequence (484 amino acids) did not contain a signal peptide, according to the SignalP v5.0 analysis, indicating that the protein is expressed intracellularly.

The deduced amino acid sequence of SulA1 shared 28.5% sequence identity with SulA2 and was encoded in the same gene cluster. BlastP analysis of SulA1 against the PDB database revealed that the closest related structure determined candidates shared relatively low sequence homology with SulA1, but the query coverage was high (Table 1). The closest matches were enzymes from *Bacteroides thetaiotaomicron* VPI-5482 (PDB 5G2V), *Pseudoalteromonas* sp. (PDB 6B0K), *Bacteroides fragilis* YCH46 (PDB 2QZU), *Pseudoalteromonas fuliginea* (PDB 6BIA), and *Hungatella hathewayi* (PDB 6UST). Nevertheless, analysis of the deduced amino acid sequence of SulA1 in Interpro (https://www.ebi.ac.uk/interpro/ (accessed on 25 January 2024)) confirmed some conserved motifs and residues throughout the aligned sequences (Figure 2). All sequences contained the FGly-SULF signature motif C/S-X-P-X-R, essential for the formation of the FGly residue, and the two catalytic residues (Cys57 (converted to FGly) and His190). In addition, four residues were identified as being calcium-binding (Asp17, Asp18, Asp277, and Asn278), and they were conserved in the enzymes and located close to the active site.

The high query coverage (Table 1) indicated an overall conserved topology. However, it should be noted that aligning SulA1 with the potential sulfatase SulA2, illustrated within the cluster in Figure 1, as well as with two recently characterized GalNAc4S-acting CS/DS sulfatases from marine *Photobacteria* sp. QA16 (GenBank UFQ91287.1 and QAB47431.1), a notably higher query coverage of 90% was observed. In contrast, the CS sulfatase that acts on GalNAc6S, from another *Photobacterium* strain (GenBank UFQ91288.1), showed a lower coverage of 67%. As observed for the PDB sequences (Table 1), sequence identity remained low, ranging from 26.6 to 28.01%. Interestingly, the SulA2 demonstrated higher query coverage with the CS sulfatase acting on GalNAc6S (GenBank UFQ91288.1, 96% coverage). The SulA2 sequence contained the FGly-SULF signature motif, two catalytic residues, and four calcium-binding residues, as observed for the other sulfatase amino acid sequences.

To further investigate structure–function relationships, three-dimensional models of the SulA1 enzyme were constructed via two alternative approaches: homology-based structural modelling using YASARA and structural prediction using AlphaFold2. The respective approach resulted in a monomeric 3D molecular model, with Ca^2+^ included as a ligand.

In the homology model, five templates from PDB (Table 1) were used to obtain a hybrid model, which was assessed as “satisfactory” according to the overall Z-score (Z = −1.786) (Table 2), regardless of the low sequence similarity with the crystallographic structures.

A significant portion of the predicted three-dimensional model of the enzyme generated using AlphaFold2 was of good quality and showed a high predicted local distance difference test (pLDDT) score above 90% (Figure 3A). Low PLDDT was only observed for the loop Ala360-Val376 (ADDGTGDAAGRAENAVV) and for the C-terminal part of the model (Figure 3A).

In both cases, the modelled 3D structure revealed a typical sulfatase fold consisting of a large N-terminal domain and a smaller C-terminal domain, each one centered on a β-sheet. Detailed comparison of the models, however, revealed discrepancies, particularly in the size and orientation of loops (Figure 3B and Appendix A), and in the orientation of the active site residue Arg61 (Figure 3C). Considering the relatively low sequence identity to the structure-determined candidates (Table 1), the prediction from AlphaFold2 was used for a further evaluation.

The N-terminal domain (Domain I) encompassed the β-strands β1, β2, and β4-β8, β11, β14, and β15 surrounded by the α-helixes α1, α2, α5, α6, α7, α8, α10, and α13 (Figure 4). The C-terminal domain (Domain II) consisted of a β–sheet formed by β17–β20 strands, surrounded by α-helices α17–α21 (Figure 4). The catalytic site was located in the interdomain region, which also contained the Ca^+2^ binding site with the conserved residues Asp17, Asp18, Asp277, and Asn278 (Figure 4B).

Based on the location of *sulA1* in the genome, potential activity on CS-AC degradation products produced with ChoA1 was suspected. Hence, the sulfated monosaccharides GalNAc4S and GalNAc6S were docked into the active site (Appendix A). However, it was challenging to predict which substrate would undergo desulfation from the docking results. The sulfate groups of GalNAc4S and GalNAc6S were orientated in opposite directions in the active site in the model. The 4S-group did not interact with the calcium-binding site (Appendix A), but the binding energy was more favorable for GalNAc4S. The distance between the sulfate group and the catalytic residues was, however, longer for GalNAc4S compared to GalNAc6S (and the distance to the catalytic residues was not significantly different when changing the Cys57 into the FGly residue). However, as seen below, GalNAc4S was the only biochemically identified substrate. This indicated that fine-tuned conformational changes may be crucial for sulfatase specificity. Such conformational flexibility, which may be influenced by solvent effects, substrate dynamics, as well as specific interactions critical for catalysis, is difficult to consider in docking models. Discrepancies may also increase by the oversimplification of experimental conditions (such as pH, temperature, and ionic strength) in modeling and docking, which can significantly influence enzyme specificity in vivo [24].

### 2.2. Recombinant Production in E. coli, and Purification of SulA1

The gene designated *sulA1* was cloned and expressed in an *E. coli* system (pVK13/*E. coli* Bl21(C43)). The recombinant SulA1 (484 amino acids) was produced in a soluble form, exhibited sulfatase activity using *para*-nitrocatechol-sulfate (*p*NCS) (Table 3), and was subsequently purified for further characterization. The molecular weight of the recombinant protein was estimated with SDS-PAGE and was aligned with the calculated weight of 51.9 kDa (Figure 5). Purification of the recombinant protein was efficiently achieved using his-tag affinity beads (Figure 5A,B). The specific activity of purified SulA1 increased approximately 4-fold compared to the corresponding activity in the crude extract under standard conditions (Table 3). The SulA1 amino acid sequence was also subjected to peptide mass fingerprinting, validating 100% sequence identity in peptide fragments (generated in silico from the software MASCOT, https://www.matrixscience.com/), corresponding to 40% query coverage of the SulA1 amino acid sequence (Figure 6).

### 2.3. The pH and Temperature Profile of SulA1

The activity of SulA1 on *p*NCS was pH-sensitive, with maximum relative activity observed at pH 5.5 (Appendix A). The activity significantly reduced to 21% and 54% of its optimum at pH 5.0 and pH 6.0, respectively (Appendix A), highlighting the sensitivity of the activity to minor pH fluctuations.

The apparent optimum temperature for *p*NCS activity at pH 5.5 was observed between 40 °C and 50 °C (Appendix A), while, at 60 °C, the SulA1 had lost 83% of the activity. This temperature range was lower than the observed unfolding temperature (*T_m_*) of the enzyme, as determined with differential scanning fluorimetry (DSF), which was 67 °C at pH 6.0 and increased with increasing pH up to pH 7 with a *T_m_* = 72.4 °C (Figure 7). This showed that the substrate affinity in the active site was lost prior to the unfolding of the enzyme.

The *T_½_*, defined as the temperature at which 50% of the relative activity is lost after 30 min of incubation, was determined to be 55 °C (Appendix A). The activity was less susceptible to lower temperatures, as at room temperature approximately 50% of the optimal *p*-activity was observed (Appendix A). Analysis of kinetic parameters of SulA1 was attempted using *p*NCS under standard conditions. However, there appeared to be a linear relationship between substrate concentration and reaction rate, suggesting that there was a high *K_m_* and a low affinity of SulA1 for this substrate.

### 2.4. Storage Stability

It was noted that purified SulA1 precipitated during storage at 4 °C and −20 °C, leading to a loss of activity. Therefore, storage optimization was investigated using *p*NCS as the substrate. It was found that when the SulA1 was stored in a 100 mM sodium acetate buffer supplemented with 20% glycerol at pH 5.5, the enzymatic activity remained stable for at least 5 weeks at −80 °C. In contrast, storing purified SulA1 in the same buffer at −20 °C led to a small decrease in activity. After one week of storage, the remaining relative activity was measured to be 86%, which remained unchanged after 5 weeks of storage. Conversely, when SulA1 was stored in 100 mM sodium acetate buffer pH 5.5 at 4 °C, a substantial reduction in activity was observed. After 1 week of storage, the relative activity dropped to 26%, and after 5 weeks, it declined to 13% of the initial activity.

### 2.5. Cation Dependence

Due to the presence of ligands for a potential calcium-binding site (Figure 2 and Figure 3), the effect of mono- and divalent cations on the activity was tested (Appendix A). It was found that different concentrations of the divalent cation Ca^2+^ (ranging from 0 to 30 mM) had an impact on the activity, increasing the relative activity on *p*NCS 3.5-fold at concentrations from 15 to 30 mM CaCl_2_ (Appendix A). This fits with the presence of the conserved residues (Asp17, Asp18, Asp277, and Asn278), and corresponding residues have, in previous studies of related enzymes, been identified as ligands for calcium binding (and in a few cases magnesium binding) in structure-determined candidates [4,25]. Moreover, it was observed that EDTA, which effectively chelated the Ca^2+^ ions, decreased SulA1 activity by ~90% (Appendix A), showing that the Ca^2+^ ion is needed for the binding of the substrate in the SulA1 active site.

DSF analysis, however, showed that increasing concentrations of CaCl_2_ slightly reduced the unfolding temperature of SulA1, while increasing concentrations of EDTA did not result in any reduction of the unfolding temperature (Table 4). Thus, calcium did not play any crucial role in stabilizing the structure of SulA1 but had a large effect on substrate affinity.

### 2.6. Sulfatase Activity against Natural Substrates

To investigate the substrate specificity of SulA1, various types of CS sulfated mono-, di-, and oligo/polysaccharides, with different sulfation patterns (Table 5), were used in a substrate screening. Given SulA1′s location upstream of the chondroitin lyase, ChoA1 (Figure 1), it was reasonable to expect activity related to CS. However, activity was only confirmed in reactions containing the monomeric sulfated amino/sugar unit with sulfation on C4, i.e., *N-*acetyl-D-galactosamine-4-sulfate (GalNAc4S), derived from CS-A. This was observed as a shift of mobility in the TLC, corroborating the formation of *N-*acetyl-D-galactosamine (GalNAc). No such change was detectable for the monomeric sulfated amino/sugar unit with sulfation on C6, or on CS-disaccharides with different sulfation patterns (Δdi0S, Δdi4S and Δdi6s), respectively (Figure 8A). As indicated by the DSF data, the pH condition that conferred the highest stability to SulA1 was found near pH 7 (Figure 8). Subsequently, an investigation of the pH optimum for SulA1, utilizing GalNAc4S as the substrate, was carried out, and the outcomes were visualized through TLC analysis (Figure 9A). These findings confirmed that the optimal pH range for this substrate was higher (pH 6.0–6.5) than initially expected (pH 5.5 for *p*NCS). Consequently, the same CS sulfated mono-, di-, and oligo/polysaccharides were reevaluated as substrates, but this time at a pH of 6.5 instead. However, enzymatic activity was still only confirmed on GalNAc4S, underscoring that SulA1 was only active on the 4-*O*-sulfate ester bond on the monosaccharide. Furthermore, complementary data from HPAEC-PAD analysis confirms the production of GalNAc following incubation with SulA1. A minor peak at 41 min (Figure 8B) was observed when the GalNAc4S had not undergone SulA1 digestion. After 30 min of digestion with SulA1 a peak appeared at 9.5 min in the HPAC-PAD spectrum, indicative of the formation of GalNAc (Figure 8C). The assumption is based on control experiments where a GalNac and galactose (Gal) were separately analyzed, resulting in the detection of a peak earlier (7.75 min) than for the GalNAc standard peak (9.5 min; Appendix A). Other sulfated natural substrates, such as heparin, κ-carrageenan, and seaweed derived fucose-containing sulfated polysaccharides (FCSPs), were also investigated as substrates for SulA1, but no activity was detected. Interestingly, despite the structural similarity to D-galactose-4-sulfate (Gal-4S), which is the sulfated monosaccharide found in κ-carrageenan, SulA1 was unable to cleave the sulfate ester group from it. This suggests that SulA1 exhibited a highly specific substrate preference towards GalNAc4S, potentially requiring the amine group for precise positioning within the active site. These results strongly implied that SulA1 is involved in the complete degradation of CS-A, in combination with the previously characterized ChoA1.

Relatively high activity at both pH 5.5 and pH 6.5, investigated with TLC (Figure 9A), indicated a broader activity range for the natural substrate compared to the artificial substrate *p*NCS, which showed a narrow pH optimum profile (Appendix A).

Furthermore, the specific activity of pure SulA1 was determined to be 6.8 mU/mg SulA1 at pH 5.5 with 20 mM GalNAc4S and 14.0 mU/mg SulA1 at pH 6.5 with 15 mM GalNAc4S, (Table 6), respectively.

Activity assays using different concentrations of substrate (GalNAc4S) were carried out to investigate if the SulA1 followed Michaelis–Menten kinetics. In contrast to *p*NCS, SulA1 followed Michaelis–Menten behavior using GalNAc4S at pH 5.5 (Figure 9B). The Michaelis–Menten behavior was also observed at pH 6.5, but was not as clear. At pH 5.5, *K_m_* was determined to be 6.94 ± 0.053 using the Michaelis–Menten nonlinear fitted curve. As 16.5 µg of enzyme was used in the kinetic experiment, the Vmax at saturation corresponded to the specific activity of 6.8 mU/mg SulA1 at pH 5.5 and 40 °C.

## 3. Discussion

Certain bacteria possess remarkable abilities to efficiently utilize complex sulfated polysaccharides, including glycosaminoglycans (GAGs) such as chondroitin sulfate (CS), for their survival [26]. Consequently, there has been considerable interest in exploring modifications of sulfation patterns in CS and DS to unravel the intricate structure–function relationships associated with CS bioactivities. Despite the abundance of genomic data (with over 41,000 sulfatase entries documented in SulfAtlas), only a few sulfatases have been crystallized, and knowledge regarding substrate specificity is limited. Currently, only 11 other CS/DS sulfatases have been expressed as recombinant proteins and extensively characterized. Early research solely used synthetic aryl substrates to characterize sulfatases, such as *p*NCS, (e.g., [27,28] or *p*-nitrophenyl sulfate (*p*NP-S; [29]). Consequently, this has led to naming the enzymes as arylsulfatases instead of relating the enzymes to their natural function. A more accurate name could be related to their structural modification of the catalytic residue to formyl glycine (FGly), resulting in a structure-based classification of FGly-SULFs. Additionally, our study demonstrated that the use of *p*NCS does not necessarily correspond to the profile of the natural substrate, resulting in misleading pH information. This emphasizes the importance of identifying the enzyme’s natural substrate for a comprehensive understanding of its characteristics.

In this study, a FGly-SULF encoding gene was successfully isolated, cloned, and expressed from the marine *Arthrobacter* strain MAT3885, producing the enzyme referred to as SulA1. This sulfatase originated from a microorganism isolated with the aim to identify novel polysaccharides processing chondroitin lyases [23]. The selected sulfatase was found to be encoded in the same gene cluster as the chondroitin lyase, ChoA1, isolated in a previous work. Bioinformatic analysis revealed that the deduced amino acid sequences of the family 1 sulfatase are often similar in length, indicating a conserved topology. This observation was further strengthened by structure-based alignments and homology modeling. Notably, the conserved sulfatase Type 1 signature motif C/SXPXR was identified in SulA1 as well as in all the aligned homologous sequences, in line with the previous findings in the literature [8,9,30]. Notably, characterized CS/DS sulfatases [2,7,13] showed less than 30% identity compared to SulA1 in BLASTp analysis on NCBI, highlighting the novelty of SulA1 and the challenges in predicting substrate specificity based on its structure.

The sequence alignment and the model showed that Cys57 was the catalytic residue that required a post-translational modification to obtain the catalytically active FGly residue. Additionally, SulA1’s active site included conserved residues (Asp17, Asp18, Asp277, and Asn278), forming a metal-ion-binding site. These findings were supported by increased relative activity observed with varying concentrations of calcium. This aligns with the established understanding that FGly-SULFs typically require divalent cations, such as Ca^2+^ (and in a few cases Mg^2+^), to facilitate substrate binding and polarization [2,7,25]. However, when integrating the DSF assay with varying concentrations of calcium and EDTA, it was revealed that calcium did not constitute to any improvement in the thermostability of SulA1. This was observed by a slight decrease in *T_m_* with an increased calcium concentration and a lack of effect on *T_m_* with EDTA. Conversely, calcium enhanced catalytic activity, and may simultaneously induce conformational changes that compromise the overall stability of SulA1.

The activity of SulA1 was initially assessed in a screening trial using the synthetic substrate *p*NCS. This trial produced a measurable and colorimetric change compared to the blank control. Notably, SulA1 demonstrated sensitivity to both pH and temperature. Particularly, temperatures exceeding 40°C have been reported to significantly impact the relative activity of CS/DS sulfatases, whereas relatively high activity can be observed at room temperature [10,13]. However, the narrow pH was especially low using *p*NCS as the substrate. On the other hand, DSF data at different pH levels, along with confirmed activity on GalNAc4S, affirmed that SulA1’s working pH range is between 5.5 and 7.0. This aligns with SulA1’s characterization as a cytosolic enzyme.

Estimation of kinetic parameters using *p*NCS as the substrate was not feasible due to a linear relationship between substrate concentration and velocity. This suggested that SulA1 has a low affinity for *p*NCS. A similar linear relationship was reported in previous studies [27,31]. Thus, while *p*NCS serves as a convenient and rapid method for testing the activity of potential FGly-SULFs, it is important to note its limitations. It is not suitable for conducting comprehensive kinetic studies or detailed characterization studies. For such investigations, the identification of natural sulfated substrates is necessary. Among mono-, di-, and oligosaccharides, it was found that SulA1 showed high specific activity towards the amino monosaccharide, GalNAc4S. These findings are in line with the sulfation pattern on the substrate used to isolate the marine *Arthrobacter* strain MAT3885 and with the localization of the *sulA1* gene upstream of the gene encoding the chondroitin lyase, which was previously characterized by Kale and coworkers [23]. No activity was detected using *N*-Acetyl-D-galactosamine-6-sulfate (GalNAc6S) as the substrate. Additionally, SulA1 showed no activity towards any of the CS-oligosaccharides with C4 and/or C6 sulfation. These results suggested that SulA1 is highly specific and is likely involved in the final steps of the CS degradation pathway in a similar manner, as proposed by Wang and coworkers [6]. Furthermore, it can be hypothesized that the neighboring sulfatase (SulA2) located in the same cluster may exhibit complementing sulfatase activity, e.g., towards the GalNAc6S unit. This speculation was supported by the characterization of ChoA1 as a chondroitin AC lyase [23], exhibiting activity towards CS-4S and CS-6S. Additionally, the amino acid sequence of SulA2 shares high topology conservation with the exolytic CS *N*-acetyl-galactosamine-6-*O*-sulfatase [7], with 96% coverage, which could be indicative for functional conservation. No activity was detected for either heparin or κ-carrageenan-derived saccharides, nor for the fucose-containing sulfated polysaccharides (FCSPs) from brown seaweed. This confirms the strict substrate preference of SulA1.

The specific activity of SulA1 was determined at pH 5.5 and pH 6.5 using 20 mM and 15 mM GalNAc4S, respectively, at 40 °C for 30 min, which resulted in specific activities of 6.8 mU/mg and 14.0 mU/mg SulA1. In comparison to other CS-sulfatases acting on the GalNAc4S unit in CS [2,13], these values are low but higher than the exo-acting enzyme termed PB_3285 (3.64 mU/mg, acting on the CS disaccharide 6 sulfate). SulA1 also appeared to strictly prefer the sulfated monosaccharide unit, whereas enzymes such as the endoVBSF from *Vibrio* sp. and exoPB4SF from *Photobacterium* sp. demonstrated activity against both poly and di-saccharides of CS with 4-*O*-sulfation.

Overall, the isolation and characterization of SulA1 provide new insights into the structure, function, and substrate specificity of *N*-acetyl-D-galactosamine-4-*O*-sulfate sulfatase, which is involved in the CS metabolism in the marine *Arthrobacter* strain MAT3885. The addition of 20% glycerol to the purified SulA1 in a 100 mM sodium acetate buffer at pH 5.5, followed by storage at −80 °C, successfully prevented precipitation and maintained SulA1’s activity for a minimum of 5 weeks. This preservation of activity not only ensures reproducibility, but also reduces cost and supports its potential in industrial applications.

## 4. Materials and Methods

### 4.1. Bacterial Strain and Plasmids

The expression vector pJOE3075 [32] and the bacterial host *Escherichia coli* BL21 C43 [F^−^
*ompT hsdS*_B_ (r_B_^−^ m_B_^−^) *gal dcm* (DE3)] [33] were used for the recombinant production of the sulfatase, SulA1, from the marine *Arthrobacter* strain MAT3885, and the plasmid pJOE3075 was used as vector for the heterologous expression in *E. coli* [32].

### 4.2. Sequencing, Bioinformatic Analysis, and Structure Modeling

The genome sequencing and annotation of the marine *Arthrobacter* strain MAT3885, using the RAST Server (Rapid Annotation using Subsystem Technology) (rast.nmpdr.org/ (accessed on 2 December 2023)) [34], was performed as previously described by Kale and coworkers [23]. The annotated 16S rDNA gene was subjected to similarity analysis using BlastN to identify the phylogenetically closest related microbial strains. Potential gene clusters in the vicinity of the CS-lyase (identified by Kale et al. [23]) were analyzed based on the RAST-data, allowing us to complete identification of the gene encoding the putative sulfatase (SulA1). The presence of any potential signal peptide was analyzed with Signal IP v5.0 [35].

Sequence similarities between the deduced amino acid sequence of SulA1 and structure determined sulfatases were analyzed using BlastP against the PDB database in NCBI. Conserved residues and motifs were analyzed by submitting the sequence to Interpro (https://www.ebi.ac.uk/interpro/ (accessed on 18 February 2024)). The output was displayed in a multiple sequence alignment, with the identified FGly-SULFs amino acid sequences (detailed below) using the Clustal Omega Multiple Sequence Alignment tool at ENBl-EBI (https://www.ebi.ac.uk/ (accessed on 25 January 2024)).

The SulA1 amino acid sequence was aligned with five best matching sulfatases of known three-dimensional structures from Protein Data Bank (PDB), originating from *Bacteroides thetaiotaomicron* VPI-5482 (PDB 5G2V), *Pseudoalteromonas* sp. (PDB 6B0K), *Bacteroides fragilis* YCH46 (PDB 2QZU), *Pseudoalteromonas fuliginea* (6BIA), and *Hungatella hathewayi* (PDB 6UST). Likewise, the SulA1 amino acid sequence was aligned and used for a comparison against four characterized CS/DS sulfatases derived from *Vibrio* sp. FC509 (PDB 6J66-A), *Photobacterium* sp. (PDB QA16) (GenBank UFQ91287.1 and QAB47431.1), and from *Photobacterium* sp. (GenBank UFQ91288.1), and the SulA2 amino acid sequence found within the chondroitin lyase cluster.

Based on these data, the 3D structure of SulA1 from marine *Arthrobacter* strain MAT3885 was built using YASARA software (http://www.yasara.org/) [36] with the templates highlighted in Table 1 at default settings (Table 7) to obtain a hybrid model. The structure was analyzed and depicted with Chimera [37]. The enzyme–ligand interaction was analyzed with LigPlot+ [38]. In addition, the SulfAtlas database (https://sulfatlas.sb-roscoff.fr/ (accessed on 18 February 2024)) was consulted to confirm the novelty of SulA1.

### Substrate Docking and MD Simulation

The three-dimensional model of the enzyme was also obtained using AlphaFold2 via ColabFold v1.5.2: AlphaFold2 (colab.research.google.com/github/sokrypton/ColabFold/blob/main/AlphaFold2.ipynb (accessed on 2 February 2024) which was hosted on Colab notebook. The model confidence was assessed via the pLDDT score, and the top ranked model was used for docking experiments. The figures were illustrated using the PyMOL v2.5.4 program (Schrödinger, [39,40]).

Docking of the enzyme with the different sulfated sugars, Gal4S, GalNAc4S, and GalNAc6S, was conducted with the local docking program implemented in YASARA. The ligands were built with the oligosaccharides building tool from YASARA. Each structure was energetically minimized using the AMBER14 forcefield. The three-dimensional model of the enzyme–ligand complexes and the active sites were graphically presented using the PyMOL v2.5.4 program (Schrödinger). The model of the enzyme and the enzyme–ligand complexes were subjected to MD simulations at 293 K and pH 6.0 for 50 ns using a YASARA macro in a simulation cell 20 Å larger than the protein. Snapshots were stored every 100 ps, and the AMBER14 forcefield was used to calculate the energies.

### 4.3. Cloning of the Sulfatase Gene sulA1

The *sulA1* gene from the marine *Arthrobacter* strain (MAT3885) was amplified via PCR using forward primer SulA1-Nde-f (5′-CGAATTCCATATGGTCAGCTCGTCCCCTGCG-3′) targeting the 5′ end of the orf including the deduced ATG start codon and the reverse primer, SulA1- BamHi-r-his (5′-CGCGGATCCGATTCCATCCCAGTTCGGCG-3′), targeting the 3′ end of the orf excluding the stop codon. The PCR product was digested with *Nde*I and *Bam*HI and ligated into *Nde* I and *Bam*HI-digested pJOE3075 vector. The resulting plasmid with *sulA1* was designated pVK13. As the reverse primer did not include a stop codon, the *sulA1* gene was cloned upstream of and in frame with 6 × 3′ histidine codons.

*E. coli* BL21(C43) was transformed with the expression vector containing the inserted gene by electroporation [41]. The transformed cells were screened for the *sulA1* gene by performing PCR using the above-described primers. Correct sequence and fusion in the plasmid were verified with sequence analysis.

### 4.4. Production, Purification, and Storage of the Recombinant Sulfatase

The *E. coli* BL21(C43) strain transformed with pVK13 was inoculated in LB broth containing ampicillin (100 µg/mL) overnight at 37 °C. The overnight culture was further diluted in fresh LB-amp broth (1:100) and cultivated at 37 °C and 200 rpm until an OD_600_ between 0.7 and 0.9 was reached. Gene expression was induced with 10% (*w/v*) L-rhamnose (final concentration of 0.1%). The culture was then grown overnight at room temperature. Cells were harvested via centrifugation (35,700× *g* for 10 min at 4 °C) and resuspended in a lysis buffer and further disrupted via sonication. The cells were kept on ice and centrifugated at 16,000× *g* for 30 min at 4 °C to separate the supernatant from the insoluble debris.

Thereafter, SulA1 was purified using PureCube His Affinity MagBeads (Cube Biotech; https://cube-biotech.com/solubilization-database (accessed on 2 February 2024), Monheim, Germany) using the protocol provided by the manufacturer. In brief, protein purifications were made with PureCube 100 INDOGO Ni-MagBeads (Cube Biotech), using the recommended lysis, wash, and elution buffers. Lysis, binding, and elution buffers consisted of 50 mM NaH_2_PO_4_ and 300 mM NaCl (pH 8.0) with varying concentrations of imidazole (10, 20, and 500 mM, respectively). After purification, the elution buffer was exchanged with 100 mM acetate buffer pH 5.5 using Amicon Ultra-0.5 centrifugal filter unit with a 30 K cut-off (Merck Millipore, Darmstadt, Germany). The expression level and purification of SulA1 were verified with SDS-PAGE, as described below.

The purified SulA1 enzyme was prepared for storage experiments as follows: First, the enzyme was added to a 30 kD Amicon ^®^ 0.5 mL centrifugal filter, and the elution buffer was exchanged with 0.1M sodium acetate buffer pH 5.5, following the manufacturer’s protocol. Afterwards, the solution was supplemented with 80% glycerol to achieve a final concentration of 20%.

### 4.5. Total Protein Content and Protein Distribution

Total protein content, including the protein content in the purified SulA1 fraction, was determined with the Bradford method [42], with bovine serum albumin as standard. Furthermore, the size distribution of proteins produced in all expression trials and the purity of SulA1 were displayed using sodium dodecyl-sulfate polyacrylamide gel electrophoresis (SDS-PAGE), using Mini-PROTEAN^®^ TGX Stain-Free^TM^ Precast Gels (4–20%, 15–well), in a Tris-glycine-SDS (TGS) running buffer.

### 4.6. Sulfatase Activity—Standard Assay

To investigate the sulfatase activity, a standard assay based on the synthetic substrate *p*-nitrocatechol sulfate dipotassium salt (*p*NCS; 2.05 mg/mL; Merck, Darmstadt, Germany) was used. In short, 5 µL enzyme extract was mixed with 20 µL 6.5 mM substrate, 10 µL 100 mM acetate buffer, pH 5.5, and 15 µL water, and the reaction was carried out at 40 °C 30 min. The reaction was stopped by adding 100 µL 1 M NaOH. The activity was determined spectrophotometrically at 515 nm. All activity tests on *p*NCS were performed in triplicate and expressed as mean ± SD.

### 4.7. Temperature Optimum, Thermal Deactivation, and Storage Capacity

The optimal temperature was determined using *p*NCS (see protocol above) assayed at varying temperatures between 20 °C and 90 °C for 30 min.

Thermal deactivation was determined as half-life (*T_½_*), defined as the temperature at which the enzyme lost 50% of its activity. This was completed by pre-incubating the enzyme at various temperatures (25 °C, 30 °C, 40 °C, 50 °C, 60 °C, and 70 °C) for 30 min, followed by the *p*NCS activity assay under optimal conditions (pH 5.5 and at 40 °C) for 30 min. Residual activity was calculated and plotted against the temperature.

The storage capacity of purified SulA1 in 20% (*v*/*v*) glycerol was tested at three different temperatures: −80 °C, −20 °C, and 4 °C, respectively. The activity was measured by using the standard assay (see above) before storage, and after 1 and 5 weeks of storage, respectively, for each storage condition.

### 4.8. pH Optimum

The sulfatase activity against *p*NCS was measured at pH ranging from 4.0 to 9.0 by using 100 mM acetate buffers (4.0 to 6.5) and 100 mM phosphate buffers (7.0 to 9.0). Reactions were conducted at 40 °C for 30 min. Similarly, the pH optimum for SulA1 was investigated using GalNAc4S as the substrate. In short, 20 µL GalNAc4S (2 mg/mL) was mixed with 15 µL pure SulA1 (16.5 µg), 10 µL 100 mM acetate buffer (ranging from 5.0 to 6.5) or 10 µL 100 mM phosphate buffers (7.0 to 9.0), and 5 µL Milli-Q water. Reactions were carried out at 40 °C for 30 min and further investigated with TLC (thin-layer chromatography; see protocol below).

### 4.9. The Effect of Cations and EDTA on Activity

To screen for the effect of different cations effect on SulA1 activity stock solutions of 100 mM MgCl_2_, CaCl_2_, KCl, and NaCl were prepared in Milli-Q water and diluted to different concentrations (20 mM, 40 mM, 60 mM, 80 mM, and 100 mM). EDTA was made at 10 mM concentration. Purified SulA1 (3.3 mg/mL) was diluted 3× (1.10 mg/mL), whereas 5 µL SulA1 was pre-incubated for 2 min at 37 °C with 10 µL of 100 mM acetate buffer, pH 5.5, and 15 µL of one of the cation’s solutions (at the concentrations defined above), leading to the final concentrations in the assay to become 6 mM, 12 mM, 18 mM, 24 mM, and 30 mM. Moreover, 20 µL substrate *p*NCS (2.05 mg/mL)) was added, and the enzyme activity was assayed at optimum conditions. Given the significant impact of calcium ions, a comprehensive evaluation of calcium concentrations was conducted. The tested concentrations included 0 mM, 5 mM, 10 mM, 20 mM, 40 mM, 60 mM, 70 mM, 80 mM, and 100 mM, corresponding to the final concentrations in the assay, as follows: 0.0 mM, 1.5 mM, 3.0 mM, 6mM, 12 mM, 18.0 mM, 21.0 mM, 24.0 mM, 27.0 mM, and 30 mM.

### 4.10. Differential Scanning Fluorimetry (DSF) in Presence and Absence of Calcium Ions and EDTA

Thermal stability SulA1 (0.56 g/L in McIlvaine buffer system) was examined under various pH conditions (ranging from 4 to 8.3) using differential scanning fluorimetry (DSF), conducted with the Prometheus NT 48 nanoDSF instrument (NanoTemper Technologies, GmbH, Munich Germany).

The effect of CaCl_2_ concentrations on the SulA1 thermostability was investigated in the same instrument using 50 µL 100 mM citrate phosphate, buffer (pH 6.03) with different concentrations of CaCl_2_ (0, 7, and 14 mM, at pH 6.03). In addition, all Ca-ions were chelated using EDTA concentrations from 0 to 5 mM. For each experiment, 10 µL of each sample was loaded in the instrument capillaries subjected to a temperature ramp from 20 °C to 90 °C, with a gradual increase of 1 °C per min. To monitor the unfolding of the protein intrinsic fluorescence at emission, wavelengths of 330 and 350 nm were used.

### 4.11. Peptide Mass Fingerprinting

Peptide mass fingerprinting was performed using ESI-Orbitrap MS/MS and MASCOT [43] to validate the amino acid sequence of the purified SulA1. The gel fragment with the correct molecular weight was cut from an SDS-PAGE and digested with trypsin in-gel [44]. After in-gel digestion, peptide fragments were determined using MS/MS analysis and compared with the deduced amino acid sequence.

### 4.12. Preparation of Sulfated Mono-, Di-, and Oligosaccharide Solutions from Various Resources

Different sulfated mono-, di-, and oligosaccharides were tested as substrates for SulA1: chondroitin sulfate dp4 (1 mg; in reaction 0.4 mg/µL, shark origin; Iduron, Alderley Edge, United Kingdom), three chondroitin-based disaccharides with different sulfation patterns; chondroitin disaccharide sodium salts without sulfation (Δdi0S), with sulfation on C4 on the *N*-Acetyl-D-galactosamine (GalNAc) unit (Δdi4S) and with sulfation of C6 on the GalNAc unit (Δdi6) (all 5 mg; in reaction 2 mg/µL; and from Dextra laboratories Ltd, Reading, United Kingdom.), the major monosaccharides units from chondroitin *N*-Acetyl-D-galactosamine-4-sulfate (GalNAc4S) sodium salt and *N*-Acetyl-D-galactosamine-6-sulfate sodium (GalNAc6S) salt (both from Dextra laboratories Ltd.; 5 mg and 1 mg, respectively; in reaction 2 mg/µL and 0.4 mg/µL, respectively), a heparin disaccharide I-H sodium salt (1mg; in reaction 0.4 mg/µL; Sigma-Aldrich), and two sulfated monosaccharides; *N*-Acetyl-D-glucosamine 6-sulfate sodium salt (major unit from heparin sulfate; 25 mg; in reaction 10 mg/µL; Sigma-Aldrich), D-galactose 4-sulfate sodium salt (the sulfated monosaccharide from κ carrageenan; 1 mg; in reaction 0.4 mg/µL; Dextra laboratories Ltd.), two κ carrageenan oligosaccharides; Neocarrabiose-4-*O*-sulfate sodium salt (25 mg; in reaction 10 mg/µL) and Neocarratetraose-41,43-di-*O*-sulfate sodium salt (5 mg; in reaction 2 mg/µL), both from Dextra laboratories Ltd., and finally two seaweed related sugars the L-fucose oligomers from *Chorda filum* cut-off <10 kDa (Fuc310; 6.25 mg/mL, in reaction 2.5 mg/µL), a sulfated polysaccharide, a Fucoidan sample from ProSea (Batch no. HFI206002, Fmc BioPolymer, 12.5 mg/mL, in reaction 5 mg/µL). Each was mixed with 15 µL 1/3 diluted pure SulA1 (1.1 mg/mL; 0.0165 mg in reaction), 10 µL 100 mM acetate buffer, pH 5.5, and 5 µL Milli-Q water. Enzyme activity was assayed after 30 min, at 40 °C. As controls, for each reaction, 20 µL of the substrate was mixed with 10 µL 100 mM acetate buffer pH 5.5 and 20 µL Milli-Q water without the addition of SulA1.

### 4.13. Analysis of the Degradation Product by Thin-Layer Chromatography

To verify the activity of SulA1 on the different substrates, TLCs were generated. Samples (2 µL) were taken from each reaction and spotted near the bottom of the TLC Silica gel 60 F_254_ (10 × 20; Merck, Darmstadt, Germany). As the mobile phase, a mixture of 1-butanol, acetic acid, and Milli-Q water (2:1:1) was run for 2 h. Thereafter, the TLC plate was dried and developed using a mixture of diphenylamine–aniline–phosphoric acid in acetone in accordance with the published procedure [45]. Then, the plates were dried and heated at 120 °C, using the TLC Plate Heater III (CAMAG^®^) until bands became visible.

### 4.14. Determination of the Kinetic Properties and the Specific Activity of SulA1

The kinetic parameters of SulA1 were attempted to be estimated by quantifying the amount of GalNAc formed after 30 min of incubation at 40 °C with different molarities (0–20 mM) of GalNAc4S, using purified SulA1 (16.5 µg). Similarly, specific activity was determined by quantifying the amount of GalNAc formed after 30 min incubation at 40 °C using 20 mM GalNAc4S, with purified SulA1 (16.5 µg). The reactions were analyzed on a Dionex ICS 5000+ (Thermo Scientific, Waltham, Massachusetts) using high-performance anion exchange chromatography with pulsed amperometric detection (HPAEC-PAD). The column used was the Dionex CarboPac 20 analytical column (3 × 150 mm) and guard column (Thermos fisher scientific, USA). Prior to analysis, the reaction mixtures were filtered through 0.45 µm syringe filter and diluted 4–60-fold in MiliQ water. A standard solution containing GalNAc ranging from 3 mg/L to 53 mg/L was prepared. For the analysis, 10 µL reaction mixture or standard was injected into the system under isocratic conditions at a flow rate of 0.5 mL/min. Both the column and compartments were maintained at a constant temperature of 30 °C. The eluents were (A) 200 mM sodium hydroxide and (C) MiliQ water; to equilibrate the system, 5.0% (A) and 95.0% (C) concentrations were used. The samples were eluted after injection under the same conditions for 14 min. Subsequently, the column was washed with 100% (A) for 7 min.

The elution peak of GalNAc4S was investigated using the same system as above but with a gradient elution protocol. Eluent composition was A: 200 mM NaOH, B: 1 M Sodium Acetate in 200 mM NaOH, and C: Milli-Q water, and the flow rate was 0.4 mL/min. To balance the system, 1.2% (A) and 98.8% (C) concentrations were used. The sample was eluted as follows: 0–20 min 1.2% (A) and 98.8% (C); 18-30 min 50% (A) and 50% (C); 30–36.1 min 20% (A); 30% (B); and 50% (C). Subsequently, the column was washed with 50% (B) and 50% (C) for ~14 min.

The experiment was performed in duplicate, and mean values along with the standard deviations were calculated, as well as specific activity. Likewise, the reaction rates were plotted against the substrate concentrations of GalNAc4S in an attempt to determine the catalytic value *K_m_* using Michaelis–Menten nonlinear equation in GraphPad Prism (10.1.2 (324)).

### 4.15. Graphs and Statistics

All graphs were generated using GraphPad Prism 10.1.2 (324). All activity tests as well as DFS were performed in triplicate and data were displayed as mean ± SD. The specific activity experiment was completed in duplicate and displayed as mean ± SD.

## 5. Conclusions

This study included the characterization of a novel sulfatase, SulA1, derived from the marine Arthrobacter strain MAT3885. Through bioinformatic analyses and comprehensive structural insights from 3D modeling, SulA1 was firmly established as a member of the sulfatase family 1. The successful expression and purification of the recombinant SulA1 facilitated a detailed examination of its biochemical properties. While SulA1 was partly characterized using the synthetic substrate *p*NCS, limitations were encountered regarding pH sensitivity and the determination of the kinetic parameters.

The investigation into natural substrate specificity revealed that SulA1 exhibited high activity against the CS monosaccharide GalNAc4S, underscoring its pivotal role in chondroitin sulfate-A degradation. Notably, the pH range was found to be broader than initially anticipated using *p*NCS as the substrate, and SulA1 displayed Michaelis–Menten behavior at pH 5.5 when GalNAc4s was used as the substrate. These findings highlight the importance of exploring natural substrates to discover the relevant conditions for enzyme catalysis. In essence, this research enhances our understanding of the catalytic mechanism of CS/DS sulfatases, providing valuable insights into their functional roles and paving the way for further investigations in this field.

## Figures and Tables

**Figure 1 marinedrugs-22-00104-f001:**
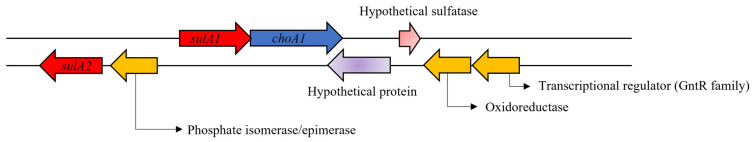
Schematic representation of the chondroitin lyase gene cluster with annotations. Within the cluster, two sulfatase-encoding genes (highlighted in red), *sulA1* and *sulA2*, were identified. The *sulA1* gene was located upstream of the chondroitin lyase gene, *choA1* (marked in blue), while *sulA2* was positioned downstream. An adjacent gene was predicted to encode a sulfatase (shaded red). Additionally, other genes within the cluster encoded proteins implicated in sugar metabolism (marked in yellow). Furthermore, a hypothetical protein (shaded purple) was found within the cluster.

**Figure 2 marinedrugs-22-00104-f002:**
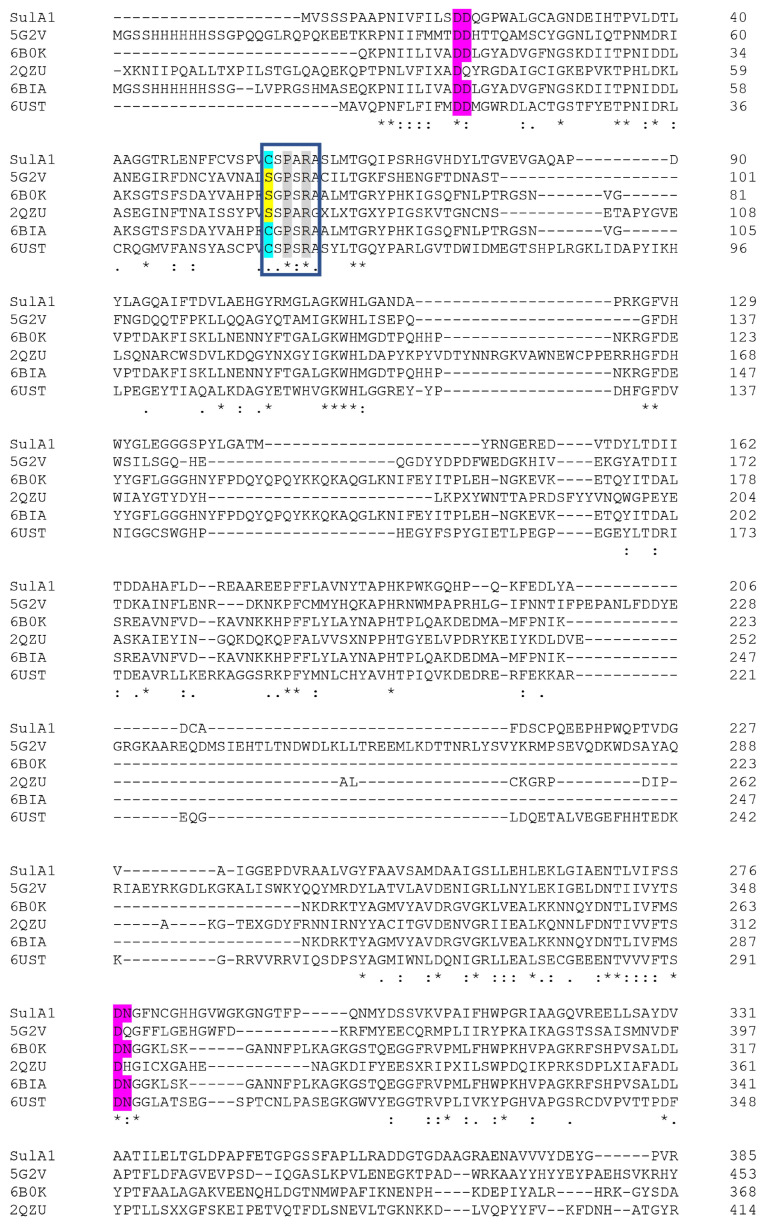
Multiple sequence alignment of SulA1 with characterized sulfatases, from *Bacteroides theraiotaomicron* VPI-5482 (PDB 5G2V), *Pseudoalteromonas* sp. (PDB 6B0K), *Bacteroides fragilis* YCH46 (PDB 2QZU), *Pseudoalteromonas fuliginea* (PDB 6BIA), and *Hungatella hathewayi* (PDB 6UST). Catalytic residues are marked in blue/yellow; the sulfatase signature motif (C/S-X-P-X-R) is marked within gray and is boxed; and Ca^2+^ binding residues are marked in pink. (*) denotes identical amino acid, (:) denotes conservative or amino acid with similar biochemical property, and (.) denotes semiconservative amino acid. The alignment was made using Clustal Omega (CLUSTAL O (1.2.4) at EMBL-EBI (https://www.ebi.ac.uk/ (accessed on 25 January 2024))).

**Figure 3 marinedrugs-22-00104-f003:**
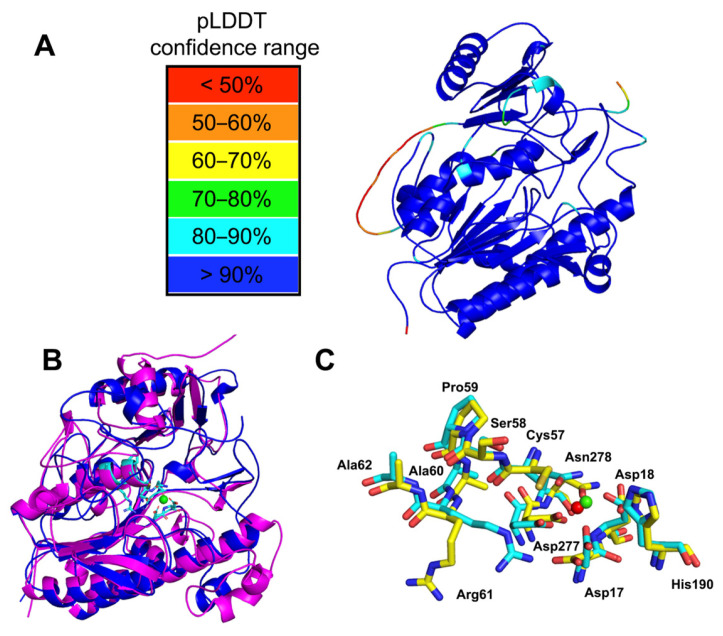
Predicted molecular structure of the sulfatase SulA1 from *Arthrobacter* strain MAT3885. (**A**) SulA1 modeled by AlphaFold2 with pLDDT confidence levels mapped onto the structure. (**B**) Comparison of the generated YASARA (Magenta) and AlphaFold2 (blue) models of SulA1; the active site residues are shown in stick and the Ca^2+^ is shown in green. (**C**) Comparison of the active site residues of the models generated using YASARA (yellow sticks, Ca^2+^ in red) and AlphaFold2 (blue sticks, Ca^2+^ in green).

**Figure 4 marinedrugs-22-00104-f004:**
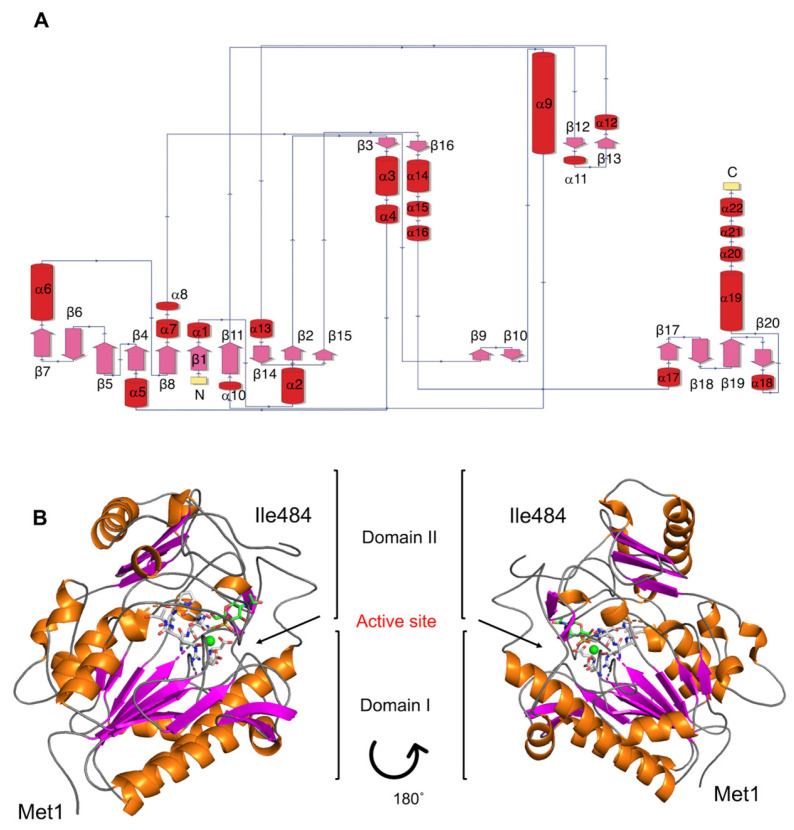
Predicted molecular structure of the sulfatase SulA1 from *Arthrobacter* strain MAT3885. (**A**) Topology map. Purple arrows represent β-sheets, while α-helices are indicated with red cylinders. (**B**) Overall structure. The ligand, highlighted in the active site in B, is N-Acetyl-D-galactosamine-4S. The green sphere represents a Ca^2+^ ion. The structure is predicted using AlphaFold2.

**Figure 5 marinedrugs-22-00104-f005:**
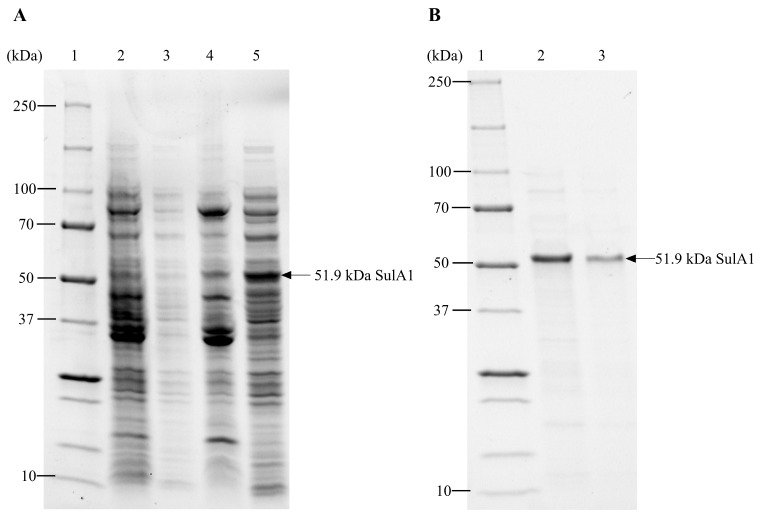
SDS-PAGE analysis of *SulA1* gene overexpression in *E. coli* BL21(C43). Gene expression was induced with 10% (*w/v*) L-rhamnose when the OD_600_ at 37 °C was between 0.8 and 0.9, and the culture was grown overnight at room temperature. Proteins were separated into a 4–20% Mini-PROTEAN^®^ TGX Stain-Free^TM^ Precast gel. Both pellets (insoluble protein) and supernatants (soluble proteins) are shown. (**A**) Lane 1, Precision Plus Protein^TM^ Standards. Lane 2, uninduced *E. coli* BL21(C43) culture, pellet. Lane 3, uninduced *E. coli* BL21(C43) culture, supernatant. Lane 4, L-rhamnose-induced *E. coli* BL21(C43) culture, pellet. Lane 5, L-rhamnose-induced *E. coli* BL21(C43) culture, supernatant. (**B**) Lane 1, Precision Plus Protein^TM^ Standards. Lane 2, first elution from PureCube INDIGO Ni-MagBead purification. Lane 3, second elution from PureCube INDIGO Ni-MagBead purification. The SulA1 protein is marked with an arrow at 51.9 kDa.

**Figure 6 marinedrugs-22-00104-f006:**
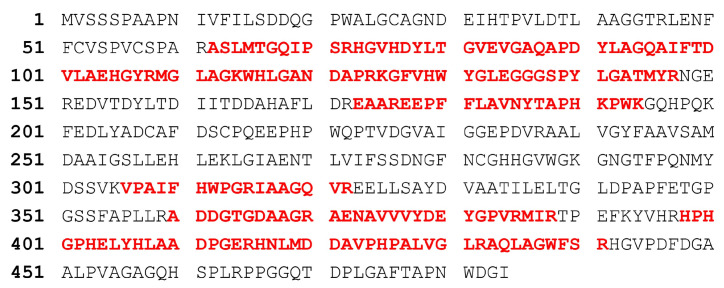
Peptide fingerprint mapping of the purified SulA1. Matched peptides are shown in bold red. The amino acid sequence coverage was 40%.

**Figure 7 marinedrugs-22-00104-f007:**
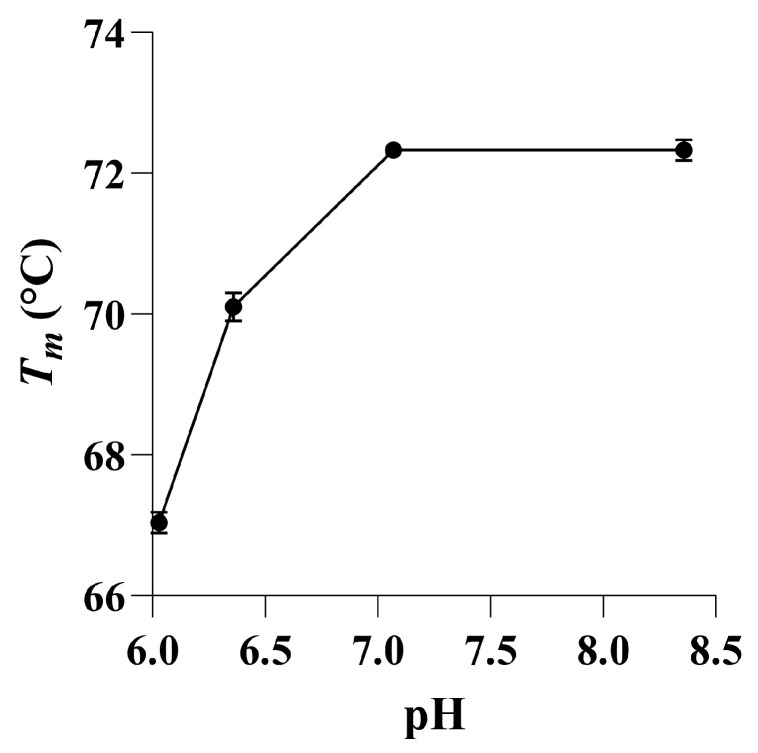
The effect of different pH on the unfolding temperature (*T_m_*) using differential scanning fluorimetry (DSF) analysis. The highest thermostability was observed at near-neutral pH conditions, resulting in a maximum *T_m_* = 72.4 °C. Scatters were unclear at a pH below 5. Data are expressed as mean ± SD of three measurements for each pH.

**Figure 8 marinedrugs-22-00104-f008:**
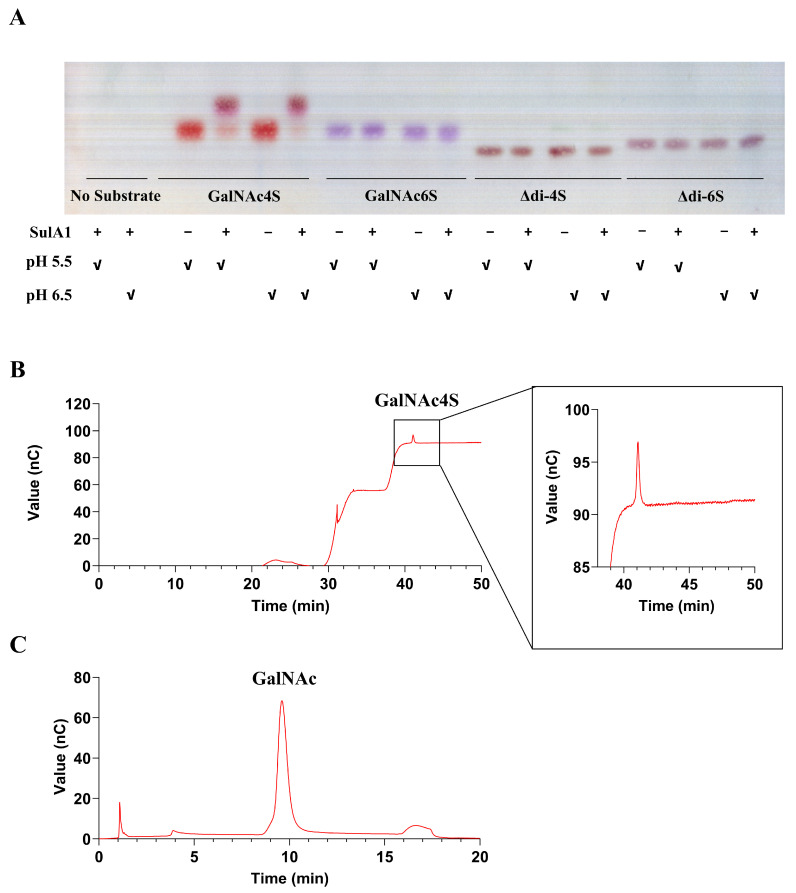
The impact of SulA1 on different chondroitin sulfate products analyzed using TLC and HPAEC-PAD. (**A**) Thin-layer chromatography (TLC) of reaction mixtures containing the monosaccharides *N*-Acetyl-D-galactosamine-4S (GalNAc4S; lane 3–6) and *N*-Acetyl-D-galactosamine-6S (GalNAc6S; lane 7–10) from chondroitin sulfate (CS) and CS-disaccharide; Δdi-4S (lane 11–14) and Δdi-6S (lane 15–18), incubate with (+) or without (−) SulA1 for 30 min at pH 5.5 or pH 6.5 indicated with ✓ using 100 mM acetate buffers. Lane 1 and 2 serve as controls where SulA1 was incubated in absence of a substrate at pH 5.5 or pH 6.5. The reactions were carried out as described in the Materials and Methods section. The mobile phase comprised 1-butanol, acetic acid, and Milli-Q water (2:1:1) and was run for 1.5 h. The TLC plate was dried and developed using a mixture of diphenylamine–aniline–phosphoric acid in acetone. (**B**) Representative chromatograms of the substrate GalNAc4S including a magnified view of the elution peak at 41 min, and the resulting formation of GalNAc (Elution peak 9.5 min; (**C**)) generated from GalNAc4S following a 30 min reaction with SulA1 and detected with HPAEC-PAD.

**Figure 9 marinedrugs-22-00104-f009:**
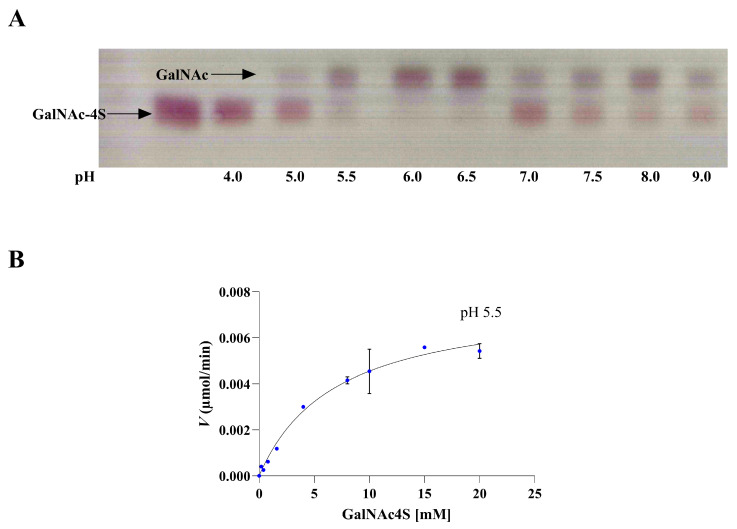
pH optimum and Michaelis–Menten curve of SulA1 using the substrate *N*-Acetyl-D-galactosamine-4-sulfate (GalNAc4S). (**A**) TLC showing reaction mixtures of SulA1 acting on GalNAc- 4S at varying pH using 100 mM acetate buffer for pH 4.0-6.5 and 100 mM phosphate buffer for pH 7.0-9.0. Reactions was assayed for 30 min at 40°C and stopped by heating the reaction to 95 °C for 3 min. The mobile phase was 1-butanol, acetic acid, and Milli-Q water (2:1:1), and the plates were run for 1.5 h. Thereafter, the TLC plates were dried and developed using a mixture of diphenylamine–aniline–phosphoric acid in acetone. (**B**) The reaction rate *V* (µmol/min) was plotted against the concentration of GalNAc4S (mM) at pH 5.5. The graph shows the GalNAc formation after the SulA1 reaction with Michaelis–Menten nonlinear curve fitting (black line); data are represented as mean ± SD of two individual experiments.

**Table 1 marinedrugs-22-00104-t001:** Crystallographic structures used as templates for the modeling SulA1.

Template (PDB Code)	Source	Sequence Coverage (%)	Amino Acid Sequence Identity (%)	Transferred Region to the Model (Residue Numbering)
5G2V	*Bacteroides thetaiotaomicron* VPI-5482	76	32.09	1–459
6B0K	*Pseudoalteromonas* sp.	89	27.27	168–180, 435–483
2QZU	*Bacteroides fragilis* YCH46	90	24.79	1–9, 283–282, 433–441
6BIA	*Pseudoalteromonas fuliginea*	90	27.20	452–484
6UST	*Hungatella hathewayi*	94	30.60	86–89

**Table 2 marinedrugs-22-00104-t002:** Hybrid model assessment according to Z-scores obtained in the YASARA program. The Z-score indicates how many standard deviations the model quality is away from the average high-resolution X-ray structure, where higher values are better, and negative values indicate a model worse than a high-resolution X-ray structure. Despite the low value for the Packaging 3D, the overall model was acceptable.

Check Type	Quality Z-Score	Comment
Dihedrals	−0.660	Good
Packing 1D	−1.453	Satisfactory
Packing 3D	−2.417	Poor
Overall	−1.786	Satisfactory

**Table 3 marinedrugs-22-00104-t003:** Relative activity of crude SulA1 and purified SulA1 tested on *p*NCS.

Sample	Relative Activity (%)
Crude SulA1	24.4 ± 1.08
Pure SulA1	100.0 ± 0.0

Data are expressed as mean ± SD of two independent reactions.

**Table 4 marinedrugs-22-00104-t004:** The effect of different concentrations of CaCl_2_ or EDTA on the unfolding temperature (*T_m_*). A decrease in *T_m_* suggests destabilization, and an increase in *T_m_* is interpreted as an increase in protein thermostability.

CaCl_2_ (mM)	*T_m_* (°C)	EDTA (mM)	*T_m_* (°C)
0	67.2 ± 0.15	0	67.5 ± 0.09
7	64.0 ± 0.06	0.5	67.3 ± 0.21
14	62.3 ± 0.10	1	67.3 ± 0.67
		2	67.1 ± 0.54
		5	67.0 ± 0.15

**Table 5 marinedrugs-22-00104-t005:** Overview of SulA1’s effect on different substrates.

Substrates	Activity

Chondroitin sulfate AC dp4 (contain both 4S and 6S)	−
Chondroitin without sulfation (Δdi0S)	−
Chondroitin sulfate (Δdi4S)	−
Chondroitin sulfate (Δdi6S)	−
*N*-Acetyl-D-galactosamine-4-sulfate (GalNAc4S)	+
*N*-Acetyl-D-galactosamine-6-sulfate (GalNAc6S)	−

Heparin disaccharide I-H	−
*N*-Acetyl-D-glucosamine-6-sulfate (GlcNac-6S)	−

D-galactose-4-sulfate (Gal-4S)	−
Neocarrabiose-4-*O*-sulfate	−
Neocarratetraose-41,43-di-*O*-sulfate	−

L-fucose from Chorda filum cut-off <10 kDa Fuc310)	−
Fucoidan	−

+ and − indicate if activity has been observed or not in the TLC plate, respectively.

**Table 6 marinedrugs-22-00104-t006:** Comparison of properties of recombinant SulA1 with selected sulfatases.

Enzyme	Type	Size (kDa)	pH Optimum	Temperature Optimum	Ligand	Substrate	Specific Activity	Reference
SulA1	*N*-acetyl-galactosamine-4-*O*-sulfatase	51.9	6.0–6.5	40–50 °C	Ca^2+^	GalNAc4S	pH 5.5 6.8 mU/mg (at 20 mM GalNAc4S) pH 6.5 14.0 mU/mg (at 15 mM GalNAc4S)	This work
endoVB4SF (*Vibrio* sp. FC509)–pdb:6J66_A	CS/DS 4-*O*-endosulfatase	59.5	8.0	30 °C	none	ΔA ^1^ CS-A	5020 and 820 mU/mg	[13]
exoPB4SF (*Photobacterium* sp.)–QAB47431.1	Exolytic *N*-acetyl-galactosamine-4-*O*-sulfatase	56.1	8.0	30 °C	Na^+^, Li^+^, K^+^, Mg^2+^, and Ca^2+^	ΔA	2857 mU/mg (for the purified protein)	[2]
PB_3262 (*Photobacterium* sp. QA16)–UF91287.1	Endo-CS-4-*O*-Sulfatase	56.7	7.0	50 °C	Ca^2+^ Ba^2+^	ΔA	165.750 U/mg	[7]
PB_3285 (*Photobacterium* sp. QA16)–UF91288.1	Exolytic CS *N*-acetyl-galactosamine-6-*O*-sulfatase	57.3	7.0	30 °C	Ca^2+^ K^+^	ΔC ^2^	3.64 mU/mg	[7]

CS: chondroitin sulfate. DS: dermatan sulfate. ^1^ ΔA CS disaccharides ΔHexUA1-3GalNAc4S. ^2^ ΔC CS disaccharide ΔHexUA1–3GalNAc6S.

**Table 7 marinedrugs-22-00104-t007:** Homology modeling settings.

Parameter	Description
Modeling speed	Slow
PSI-BLAST iterations in template search	3
Maximum allowed (PSI-)BLAST E-value to consider template (EValue Max)	0.5
Maximum number of templates to be used	5
Maximum number of templates with same sequence	1
Maximum oligomerization state	4
Maximum number of alignment variations per template	5
Maximum number of conformations tried per loop	50
Maximum number of residues added to the termini	10

## Data Availability

The data presented in this study are available upon request from the corresponding author.

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
