# Peer review of "Cloning and Characterization of a Novel N-Acetyl-D-galactosamine-4-O-sulfate Sulfatase, SulA1, from a Marine Arthrobacter Strain"

_marinedrugs, 2024, doi:10.3390/md22030104_

Round 1

Reviewer 1 Report

Comments and Suggestions for Authors

Comments on the Quality of English Language

Past tense is strongly encouraged for manuscript writing, especially the method and result sections.

Author Response

  1. Past tense is strongly encouraged for manuscript writing, especially the method and result sections. Now, the mixed past tense and present tense appear in some areas.

Answer to 1:

Thank you for bringing this to my attention. I have reading over the manuscript several of times and believe I have change to past tense.

  1. The term of the strain in the manusript is inappropriate. a marine Arthrobacter strain à a marine-derived Arthrobacter sp. (MAT3885)à Arthroactor MAT3885, à Arthrobacter strain MAT3885 à Arthrobacter sp. MAT3885.

Answer to 2:

Thank you. I have changed the naming to be marine Arthrobacter strain (MAT3885) it should be consisting

  1. rSulA1 is suggested to represent the recombinant SulA1.

Answer to 3: I appreciate your thoughtful suggestion; however, based on the International Protein Nomenclature Guidelines, SulA1 meets the criteria for proper naming

  1. L15: “ a novel sulfatase (SulA1), encoded by the gene sulA1 is characterized” “a novel sulfatase (SulA1) encoded by the gene sulA1 was characterized”.

Answer to 4:

I have changed the sentence according to your suggestion

  1. Some sentences are too long to understand and can be simplified, eg. L26-29, L624- 646 à this is the material and methods where the natural substrates are mentioned.

Answer to 5:

I have attempted to make the sentence shorter and easier to understand. However, the line 624-646  which is section 4.11 in material an methods is difficult to shorten and elaborate since it is a summarization of the natural substrates used to investigate the substrate specificity of SulA1.

  1. In the Section 2.1 and 4.2: Owing to the phylogenetic identification and genome sequencing were done previously [Kale et al., 2015], the related contents did not need to be repeated.

Answer to 6: I understand the confusion. I have tried to make the sentence clearer (L88-L91) and L489-492).

  1. L501-502: “sp.” should not be italicized.

Answer to 7: Thank you. This have been correct according to your comment

  1. L535: “Production, purification, and storage of the recombinant sulfatase” should be section 4.4?

Answer to 8: Yes, thank you, this have been corrected.

  1. L647: The title “Thin-layer Chromatography” is suggested to be “Analysis of the Degradation Product by Thin-layer Chromatography”.

Answer to 9: Thank you for this suggestion. I have changed the title according to your suggestion.

  1. L685-692: The section of Conclusions is too simple. The main results should be listed.

Answer to 10: The conclusion has been improved. I hope it fulfill your requirements

Reviewer 2 Report

Comments and Suggestions for Authors

Duagbjerg et al are presenting a study dealing with the identification, cloning, purification and production of a marine GalNAc-4A sulfatase.

The study is well written and concise, which make the manuscript pleasant to read.If the presented results deserve to be published in marine drugs, I have two main concerns on some presented results, that should be answered before considering publication.

1- the quality of the docking is questionable, as the GalNAc4S biochemical substrate is docked in an orientation that prevents sulfate hydrolysis. Therefore, one can wonder the added value of this docking,as it is clear that the docked positions do not correspond to the observed in vitro activity. The authors should either remove this docking part, or propose a docking procedure that yields an active conformation (and would help to analyse the substrate specificity).

2- The methods to determine the catalytic constants values (kcat or Vmax, and KM) from the presented graphs (Figure 10 and S4) are not precised (Michaelis Menten plot direct fitting ? Lineveawer-Burk analysis ?). in addition,the quality of the values given is questionable, as the 2 figures are clearly not hyperboles. To overcome this, activity values at much higher substrate concentration should be determined (at least 10 times higher than the KM estimated of 6 mM), and the hyperbolic fit of the MM equation should appear instead of a line representation. Eventually, the KM values are not presented on table 6, although there is a reference of the table in the main text (l. 323).

Minor comments

Results part 2.1 : please precise and discuss thesequence homology between SulA1 and SulA2

l94 : Please precise how the absence of signal peptide was detected

p8 l201 : please precise what is pNCS at its first mentioning, as the exact nature (paranitrocathecol) is only explained in the experimental procedures part.

Figure 10 and Supplemental figure S4 : the x-axis is not correct (GalNAC-4S instead of GalNAc).

Author Response

  • the quality of the docking is questionable, as the GalNAc4S biochemical substrate is docked in an orientation that prevents sulfate hydrolysis. Therefore, one can wonder the added value of this docking,as it is clear that the docked positions do not correspond to the observed in vitro activity. The authors should either remove this docking part, or propose a docking procedure that yields an active conformation (and would help to analyse the substrate specificity).

Answer to 1: Thank you for your comment. We have dedicated efforts to improve this aspect by including a prediction of sulfatase structure using AlphaFold2. Unfortunately, this did not resolve the challenges associated with molecular docking, and the difficulty in predicting weather GalNAc4S or GalNAc6S would be desulfated, persists, despite a favorable binding energy towards GalNAc4S. We have tried to come with a reasonable explanation in the result section 2.1 from L143-L207

  • The methods to determine the catalytic constants values (kcat or Vmax, and KM) from the presented graphs (Figure 10 and S4) are not precised (Michaelis Menten plot direct fitting ? Lineveawer-Burk analysis ?).

in addition, the quality of the values given is questionable, as the 2 figures are clearly not hyperboles. To overcome this, activity values at much higher substrate concentration should be determined (at least 10 times higher than the KM estimated of 6 mM), and the hyperbolic fit of the MM equation should appear instead of a line representation.

Eventually, the KM values are not presented on table 6, although there is a reference of the table in the main text (l. 323). I’ve improved the text à no Km or Vmax in the table

Answer to 2:

Thank you for bringing up a valid point, and I truly acknowledge the addressed issue. Unfortunately, it was not feasible to include a 10 times higher data point for the kinetic study due to the considerable cost associated with the GalNAc4S substrate.

In an effort attempt to address the concern, I have made improvements to Figure 10, which is now figure 9. The updated figure 9, incorporates catalytic data from pH 5.5 with corrected axes (X= GalNAc4S [mM], Y = reaction rate V in µmol/min) instead of showing the specific activity. I have included a nonlinear fitted curve based on the Michaelis-Menten equation, which appears to be a satisfactory match for the reactions conducted at pH 5.5 because the reactions at pH 5.5 showed a clear Michaelis Menten behavior. Similarly, the datapoint for the reactions at pH 6.5 may indicate Michaelis-Menten behavior, however the regression line was not hyperbolic making it difficult to determine the kinetic parameters. For that reason, a new Km was determined at pH 5.5 and expressed in the result part as mean ± SD (L305) and are not presented in Table 6. The Km at pH 5.5 was estimated using GraphPad Prism and have been stated in the material and methods section 4.13.

Minor comments

Results part 2.1 : please precise and discuss the sequence homology between SulA1 and SulA2

Answer on this request: I incorporated the details requested in the result section, specifically highlighting the homology study between SulA1 and SulA2 starting a L104. Additionally, I have included a comprehensive comparison of SulA2 with the other characterized CS/DC presented in Table 6, similarly to what was done for SulA1. Likewise, has the discussion (L463) and the material and method sections have been updated accordingly. 

l94 : Please precise how the absence of signal peptide was detected _ The SulA1 sequence (484 amino acids) did not contain a signal peptide, indicating that the protein is expressed intracellularly – reformulated the sentence to: “The SulA1 sequence (484 amino acids) did not contain a signal peptide, according to the Signal IP v5.0 analysis, indicating that the protein is expressed intracellularly.

Answer:

Thank you for this observation. I have changed the sentence L95.

p8 l201 : please precise what is pNCS at its first mentioning, as the exact nature (paranitrocathecol) is only explained in the experimental procedures part.

Answer: Thank you. This has been improved now in L.212.

Figure 10 and Supplemental figure S4: the x-axis is not correct (GalNAC-4S instead of GalNAc).

Answer: Thank you for bringing up this matter, I truly acknowledge the comments.

I have made improvements to Figure 10 (now figure 9) and S4 (now S5). The updated figure incorporates catalytic data from only pH 5.5 with corrected axes (X= GalNAc4S [mM], Y = reaction rate V in µmol/min) instead of specific activity.

Reviewer 3 Report

Comments and Suggestions for Authors

The authors described the newly found sulfatase that is not similar to previously characterized sulfatases. Substrate specificity and linkage position specificity of the enzyme was determined using natural substrates, which is good achievement in this study. The fact that the structural prediction did not provide plausible binding mode of the substrate the authors found makes us expect that the enzyme shows unique binding mode of the substrate.

However, there are several concerns on the analyses as described below.

1. kinetics

Because kinetic parameters are important, fitting curves should be appeared as main figures. The plots should be regressed in the figures. Kinetic parameters should be shown with errors (± XXX). I cannot judge whether regression is appropriate or not using the present version.

The authors describe that Vmax is 14.0 U/mg and also describe that specific activity is 14.0 in Table 6. However, Vmax and specific activity is clearly different. Specific activity is an activity in the presence of a specific concentration of a substrate.

2. error bar

Some figures do not provide error bars. (Fig.S1, S2). Please provide error bars if the authors performed the experiments in triplicate. If not, please describe that in the experimental procedures section.

3. cation dependence (Fig. S2)

The authors do not add calcium ion in the usual assay. Calcium ion seems not to be added to SulA1 (Fig. S2, the left bar). Why does this sample show such high activity while +CaCl2 5 mM reduces activity remarkably? This is quite weird.

Other comments

D and L: Their fonts should be changed to “small capital letters” throughout the manuscript.

Fig. 1. Annotations could be indicated for three yellow arrows individually.

Fig. 3. Please describe that this structure is a predicted model in the legend.

Several “E. coli” are left as plain font.

Fig. 6. Line number 202 and A overlap.

Tm, Km, Vmax, T1/2   Their fonts should be corrected. KmVmax Tm T1/2

Line 549  Please provide pH for NaH2PO4

Line 596  cations are more appropriate than calcium ions.

There is no information on cations other than calcium. In the same section, final concentrations of calcium ions (or cations ?) seems to 0, 10, 20, 30, 40, 50 mM, which are different from the values shown in Fig.S2.

Author Response

  1. kinetics

Because kinetic parameters are important, fitting curves should be appeared as main figures. The plots should be regressed in the figures. Kinetic parameters should be shown with errors (± XXX). I cannot judge whether regression is appropriate or not using the present version.

The authors describe that Vmax is 14.0 U/mg and also describe that specific activity is 14.0 U/mg SulA1 in Table 6. However, Vmax and specific activity is clearly different. Specific activity is an activity in the presence of a specific concentration of a substrate.

Answer for 1:

Thank you for bringing up this matter, I truly acknowledge the comments.

In an effort attempt to address the concern, I have made improvements to Figure 10, which is now figure 9. The updated figure 9, incorporates catalytic data from pH 5.5 with corrected axes (X= GalNAc4S [mM], Y = reaction rate V in µmol/min) instead of showing the specific activity. I have included a nonlinear fitted curve based on the Michaelis-Menten equation, which appears to be a satisfactory match for the reactions conducted at pH 5.5 because the reactions at pH 5.5 showed a clear Michaelis Menten behavior. Similarly, the datapoint for the reactions at pH 6.5 may indicate Michaelis-Menten behavior, however the regression line was not hyperbolic making it difficult to determine the kinetic parameters. For that reason, a new Km was determined at pH 5.5 and expressed in the result part as mean ± SD (L305) and are not presented in Table 6. The Km at pH 5.5 was estimated using GraphPad Prism and have been stated in the material and methods section 4.13.

  1. error bar

Some figures do not provide error bars. (Fig.S1, S2). Please provide error bars if the authors performed the experiments in triplicate. If not, please describe that in the experimental procedures section.

Answer to 2:

I appreciate the comment. I have taken your feedback into consideration and updated all the graphs to include error bars. Additionally, I have revised the “graphs and statistics” section (4.14) to ensure that everything aligns with your request.

  1. cation dependence (Fig. S2)

The authors do not add calcium ion in the usual assay. Calcium ion seems not to be added to SulA1 (Fig. S2, the left bar). Why does this sample show such high activity while +CaCl2 5 mM reduces activity remarkably? This is quite weird

Answer to 3:

Yes, I agree, the previous results appear to be confusing, and we find it challenging to provide a clear explanation. We believe it is artifact. In light of this, we have decided to exclude the activity measurements with 5 mM ion concentrations in S2 (now S4). Moreover, the updated version of the figure includes showing error bars, change of concentrations of ions, because the previously concentration inaccurately referred to the stock solution instead of the final concentration in the assay. I have resolved the concentration in Figure S2 (now S4) accordingly. I would also like to address that the SulA1 activity without cations added (left bar), is not entirely devoid by calcium. Calcium is ubiquitous in the plastic and glassware, making complete elimination challenging.

Other comments

  • D and L: Their fonts should be changed to “small capital letters” throughout the manuscript.

Answer:

Thank you, this have been improved. However, it is not very visual due to the font type required for the manuscript.

  • 1. Annotations could be indicated for three yellow arrows individually

Answer:

Annotation has been provided in figure 1 and the figure text has been updated  

  • 3. Please describe that this structure is a predicted model in the legend.

Answer:

I Changed the figure legend to highlighted it is predicted models (Figure 3-4)

  • Several “E. coli” are left as plain font.

Answer:

Thank you, I believe all have been corrected                                                                                                       

  • 6. Line number 202 and A overlap.

Answer:

This has been correct

  • Tm, Km, Vmax, T1/2   Their fonts should be corrected. KmVmaxTm T1/2

Answer:

Thank you. This has now been correct in text and figures

  • Line 549 Please provide pH for NaH2PO4.

Answer: pH was 8.0 and is now provided in the manuscript.

  • Line 596 cations are more appropriate than calcium ions.

There is no information on cations other than calcium. In the same section, final concentrations of calcium ions (or cations?) seem to 0, 10, 20, 30, 40, 50 mM, which are different from the values shown in Fig.S2.

Answer:

Thank you for bringing this to my attention. I have revised the Material and Method section (4.8) to address the inconsistency. The concentrations in S2 (now S4) were inaccurately stated, as it referred to the stock solution instead of the final concentration in the assay, which is accurately presented in S1 (Now S3). I have resolved the concentration in figure, and updated material and method accordingly (section 4.8). Additionally, I would like to clarify that the decision to test a broader range of calcium concentrations in in S3 was based on the outcomes of the cation screen in S4.

Round 2

Reviewer 2 Report

Comments and Suggestions for Authors

The authors have answered in adequation to the comments made in the first round of revision, therefore I recommend publication.

Author Response

Than you.

I want to express my sincere gratitude for your review of my manuscript. Your insightful feedback and recommendations were valuable in enhancing the quality of the content and I'm appreciate your positive recommendation for publication.

Reviewer 3 Report

Comments and Suggestions for Authors

Many of comments are answered correctly, but several concerning points are left and revision was performed somewhat carelessly. The points are described as follows.

Kinetic parameters should be shown.

Table 6 For specific activity, substrate concentration that the reaction is performed is needed.  

Fig. 9  line in the graph should be deleted. Only fitting curve should be left.

Fig. S4. Black bars mask the error bars. Please change colors.

D and L: small capital letters should be adopted.

Fig. 1 legend, the annotation shown in the figure should be noted that they are annotation in the legend.

Please check Font and typos throughout the manuscript. 

Ca2+  uppercase

Lines 273, 275, 744 Table4 Fig7  T1/2 Tm

Line 24  not hyphen but endash,  pH5.5   pH 5.5

Line 52  What is D4,5-hexuronate?

Line 58  beta-1,4-glycosidic    (add hyphen)

Style of references should be organized.

Line 715 a novel

Fig. 3&4 title  Molecular  molecular

Author Response

I want to express my sincere gratitude for your thorough review of my manuscript. Your insightful feedback and recommendations were incredibly valuable in enhancing the quality of the content. Bellow is the answers

  1. Kinetic parameters should be shown.

Answer: Certainly, the inclusion of kinetic parameters such as Km is crucial for a comprehensive understanding of enzyme activity. For that reason I have improved the text in L351, to include a more detailed presentation of Km (represented as mean ± SD). I considered to add the Km values to table 6, however I don’t believe it is meaningful when there is no basis for comparison, with the other sulfatase represented in the table. Without a comparative dataset, then adding the Km to the table did not provide substantial information.

  1. Table 6 For specific activity, substrate concentration that the reaction is performed is needed.

Answer: I have added the molarity of GalNAc in table 6. Moreover, section 4.13 (material and methods) was updated accordingly, and the substrate concentrations were specified in the main text as well (L345, and L468).

  1. 9 line in the graph should be deleted. Only fitting curve should be left.

Answer: Thank you for this valuable comment. Figure 9 has been corrected according to your request.

  1. S4. Black bars mask the error bars. Please change colors.

Answer: Figure S4 has been changed to white error bars. Hope this is satisfactory.

  1. D and L: small capital letters should be adopted.

Answer: Thank you for your feedback. I have addressed the issues you pointed out, and all corrections have now been made. It seems that some changes were not saved in the previous submission, but I have ensured that everything is in order now.

Regarding the font style for small capital letters, I have applied the small cap style in Word, as you suggested. Additionally, I have adjusted the font size slightly smaller to align with your specified requirements.

  1. Fig 1 legend, the annotation shown in the figure should be noted that they are annotation in the legend.

Answer: The figure legend has been updated.

  1. Please check Font and typos throughout the manuscript. 

Ca2+  uppercase

Lines 273, 275, 744 Table4 Fig7  T1/2 Tm

Line 24 not hyphen but –, pH5.5  à pH 5.5

 Answer: I believe that everything has been corrected.

  1. Line 52  What is D4,5-hexuronate?

Answer: this should of course be delta- 4,5-hexuronate, (now corrected), and it is referring to the unsaturated hexuronic acid formed when a GAG lyases cleaves the β1-4 glycosidic linkage between GlcNAc/GalNAc and GlcA/IdoA.

  1. Line 58  beta-1,4-glycosidic    (add hyphen)

Answer: Hyphen was added

  1. Style of references should be organized.

Answer: Thank you. This has been improved.

  1. Line 715 a novel.

Answer: Thank you for this observation.

  1. 3&4 title Molecular to molecular 

Answer: Both have been improved